# Ground Penetrating Radar as a Functional Tool to Outline the Presence of Buried Waste: A Case Study in South Italy

**Carmine Massarelli** *, **Claudia Campanale** and **Vito Felice Uricchio**

National Research Council, Water Research Institute, Via De Blasio, 70132 Bari, Italy;
claudia.campanale@ba.irsa.cnr.it (C.C.); vito.uricchio@ba.irsa.cnr.it (V.F.U.)
* Correspondence: carmine.massarelli@ba.irsa.cnr.it

**Abstract:** The ability of the ground penetrating radar (GPR) method as a rapid preliminary survey to detect the presence of illegally buried waste is presented in this paper. The test site is located in the countryside of "Sannicandro di Bari" (Southern Italy) and has a surface area of 1500 m$^2$. A total of five parallel profiles were acquired in 2014 using a geophysical survey system instrument (GSSI) equipped with 400 and 200 MHz antennae in the monostatic configuration. Two of the five profiles were registered in a control area to compare a natural condition to a suspected waste buried zone. As a result of a processing and elaboration workflow, GPR investigations allowed us to interpret the signal qualitatively within a maximum depth of about 3 m, identifying many signal anomalies, whose characteristics can be considered typical of buried waste. The GPR response of the three profiles acquired in the suspected area showed substantial differences not found in the control's profiles. Anomalies related to the presence of intense scattering, of dome structures not attributable to cavities, but rather to a flattening and compacting of different layers, therefore, less electrically conductive, were identified in the suspected area. The interpretation of the results obtained by the GPR profiles was confirmed by excavations carried out with bulldozers. Large quantities of solid waste illegally buried (e.g., waste deriving from construction and demolition activities, bituminous mixtures, discarded tires, glass, plastic, municipal waste) were revealed in all the sites where anomalies and non-conformities appeared compared to the control natural soil.

**Keywords:** ground-penetrating radar; GPR; geophysics; buried waste detection; South Italy





## 1. Introduction

Before applying decision-making and remedial strategies, the reclamation of a polluted site requires identifying illegal waste dumps. However, conventional surveys via field sampling do not fit with the detection of buried wastes inexpensively and expeditiously.

Lately, several geophysical methods have been adopted to discover and locate suspected buried waste sites [1–5].

For applications as dump characterization and monitoring, electrical (electrical resistivity tomography (ERT), induced polarization (IP) and self-potential (SP)) [6,7] and electromagnetic (transient electromagnetic methods (TEM), ground-penetrating radar (GPR)) [8] are the traditional surveys due to the sensitivity in conductivity contrasts of waste and leachate [9]. On the one hand, the latest techniques based on electrical surveys provide fast and cost-efficient measurements acquiring information at various lateral and vertical locations of the study site. On the other hand, electrical noise interferences caused by power lines, pipelines, buried casing, and fences can occur, especially in industrialized sites [10].

Among the electromagnetic methods, ground-penetrating radar (GPR), famous in the archaeological field [11], has been lately reported as achieving resounding success due to its versatility and to the possibility of numerous applications like the characterization of aquifer morphology [12,13], mapping artificial structures (e.g., pipes, tanks, cables) [14],

water content determination [15], monitoring hydrological processes [16], contaminant mapping [17], detecting of buried heterogeneities such as illegal dump deposit [18], or cavities [19].

In particular, the feasibility of GPR in the application of the identification and mapping of illegal dumping allows rapid and reliable site investigation of a possible waste site [20].

GPR advantages are survey speed, coverage of large survey sites [20–23], data processing and resolution [14–26], high implementation with other geophysical techniques [27], and short time of acquisition [28,29].

Even if GPR results are less accurate than other geophysical investigations and the penetration depth in conductive materials (>20 mS/m) such as silts and clays is minimal [9], this technique is a good compromise for ease of use, speed, and relatively high resolution (depending on the antennae and ground properties).

The need for security personnel and access to sites accompanied by the local authorities make the advantages of the technique a fundamental requirement for this type of application [30].

Indeed, in previous years, other authors [8,18,31,32] have exploited GPR practicality in outlining contamination areas due to solid waste pollution giving back results in terms of reflection anomalies correlated to the presence of contaminant spreading.

In light of this, as part of the investigations delegated by the District Anti-Mafia Directorate (D.D.A.) of Bari (Southern Italy), the State Forest Corps of Cassano delle Murge (a city of Bari Province), fearing the presence of possible buried waste, dangerous and not, in an area subject to landscape restrictions, asked the National Research Council–Water Research Institute in 2014 (24 April) for a preliminary investigative rapid and cost-effective method that could respond to their needs.

In this context, the goal of this paper is to present a challenging case study that demonstrates the potential of GPR on the identification of anomalies in the subsurface linked to the presence of suspected non-natural materials and waste illegally buried. The hypothesis is that illegal waste dumps consist of heterogeneous areas that reflect radargrams with many discontinuous intense scattering regions alternate to more regular reflections representing the inert material.

Notwithstanding care on GPR data interpretation should be taken especially for differentiation between natural or geologic materials and anthropic buried items, we evaluated this technique as the most suitable in terms of fast spatial coverage, flexibility of usage, and reliability of results [14] that best fit our purpose and available resources.

Afterward, the suspicious areas identified were investigated to field sampling to support and verify the given information and proceed to the site's clean up.

*Theoretical Framework and Principles of GPR (Ground Penetrating Radar)*

GPR is a non-destructive geophysical technique based on emitting, by a transmitting antenna, electromagnetic waves of known frequency (generally among 100 and 2500 MHz) into the probed material and receiving back the reflected pulses when the propagating waves encounter discontinuities by a receiving antenna [33,34].

The transmitter and a receiver are set in a stationary configuration, moving them over the surface to reveal reflections from the subsurface [35]. This configuration is depicted in Figure 1a.

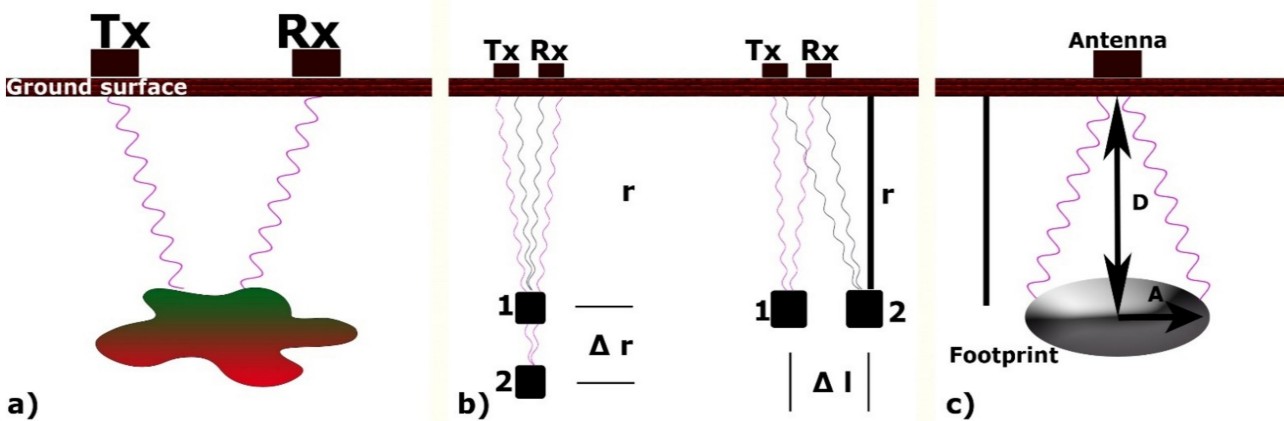

**Figure 1.** Principles of GPR (Ground Penetrating Radar). (**a**) Conceptual model of operation; (**b**) vertical and horizontal resolution can be determined by considering the response of two localized targets either in-line (left) or side-by-side (right); (**c**) simplified GPR footprint (Fresnel zone) representation in which the shaded zone depicts the area illuminated at depth from radar energy radiated from the transmitting antenna.

This type of survey aims to provide information about the subsurface with a penetration depth ranging from several tens of meters to about half a meter [36].

The propagation velocity (1) of the waves emitted by the transmitting antenna depends on the subsurface surrounding the ambience's permittivity, which is a material's ability to resist the electric field induced by the waves dependent upon the water and clay content, lithology, and bulk density of the subsurface [34,37].

$$v = c/\sqrt{k} \tag{1}$$

where

$v$ = propagation velocity of the electromagnetic wave passing through the material;
$c = 3 \times 10^8$ m/s is the speed of light in vacuum;
$k$ = relative dielectric permittivity (RDP) of the material traversed by the radar energy.

Furthermore, a material's permittivity is typically expressed as a relative quantity given by the ratio of the material's dielectric permittivity to that of free space [34].

Therefore, the reference value used in this geophysical application refers to the relative dielectric permittivity (RDP). The lower the RDP value, the greater the wave radar's velocity that propagates inside it (Table 1).

**Table 1.** Typical relative dielectric permittivity (RDP) of usual materials in applications of this type and wave propagation velocity measured at 100 MHz [33,38].

| Material | RDP | Wave Velocity (mm/ns) |
|---|---|---|
| Air | 1 | 300 |
| Dry sand | 3–5 | 120–173 |
| Clay soil (dry) | 3 | 173 |
| Asphalt | 3–5 | 134–173 |
| Quartz | 4.3 | 145 |
| Limestone | 4–8 | 100–113 |
| Granite | 5–15 | 106–120 |
| Concrete | 6 | 112 |
| Dolomite | 6.8–8 | 106–115 |
| Clay (wet) | 8–15 | 86–110 |
| Sandy soil | 10 | 95 |
| Pastoral land | 13 | 83 |
| Agricultural land | 15 | 77 |
| Freshwater | 80 | 35 |
| Seawater | >81 | 33 |

Once the propagating waves meet the first material with different dielectric properties, part of the waves is reflected while the remaining is diffused at the deeper layer [14].

The amount of the reflected wave is defined by the layer's reflection coefficient *R* expressed as [39,40]:

$$R = \left(\sqrt{K_1} - \sqrt{K_2}\right)/\left(\sqrt{K_1} + \sqrt{K_2}\right) \qquad (2)$$

with

　　*R* = reflection coefficient;
　　$K_1$ = RDP of the first layer;
　　$K_2$ = RDP of the second layer.

The electromagnetic wave absorption is more significant for higher frequencies than for lower ones, resulting in a lower penetration depth and a more accurate resolution. Conversely, smaller frequency electromagnetic waves increase the maximum penetration depth into the ground, reducing resolution [41].

Resolution can be defined as the radar system's ability to discriminate two adjacent individual targets in the subsurface separately [42]. It is differentiated, in turn, into horizontal (lateral, angular or plain) resolution ($\Delta l$) and vertical (radial, depth or longitudinal) resolution ($\Delta r$) (Figure 1c).

Horizontal resolution measures the minimum distance between two targets at the same depth to discriminate them as individual elements. It depends on the antenna's beam geometry, the type of antenna, the emitted frequencies, the target's depth, and the material's electromagnetic parameters [43]. When we set a GPR system on a surface, it looks out in a wide area under the antenna and sideways. This area irradiated by the antenna is called the Fresnel zone or antenna footprint and is an ellipse (Figure 1b) [44]. The beam can be considered as the cone of energy that illuminates the Fresnel zone [45]. A strict beam provides a better horizontal resolution.

The horizontal resolution can be approximated by calculating the Fresnel zone by Equation (3) [35]:

$$A = \lambda/4 + D/\sqrt{K-1}B = A/2D = z_0 - z_d \qquad (3)$$

where

　　*A* = major radius of the Fresnel zone;
　　*B* = minor radius of the Fresnel zone;
　　$\lambda$ = center frequency wavelength of radar energy;
　　$z_d$ = depth of the Fresnel zone in the subsurface;
　　$z_0$ = antenna elevation relative to the ground level;
　　*K* = average RDP of material from the ground surface to depth.

Vertical resolution measures the minimum distance between two vertically separated targets and provides information about the system's ability to differentiate in time two adjacent signals. It mainly depends on the radar pulse duration, which is related to the antenna's frequency. The vertical resolution achievable is influenced by the signal wavelength, and it is often considered to be between one half to one-quarter of the signal wavelength. Higher frequency signals will provide better vertical resolution [35].

To obtain a vertical resolution:

$$\Delta r \geq Wv/4 \qquad (4)$$

where

　　*W* = width at the half amplitude of the transmitted impulse (directly related to the inverse of the center frequency);
　　*v* = velocity of the electromagnetic wave that passes through the material;
　　while for a horizontal resolution:

$$\Delta l \geq \sqrt{vrW/2} \qquad (5)$$

## 2. Materials and Methods

### 2.1. Study Area and Local Geology

The GPR survey was carried out on one of the Picone Torrent slopes, in the countryside of "Sannicandro di Bari" (Bari Province), in the "Parco delle Grotte" area, a zone subjected to landscape restrictions. The area sometimes provides steep walls and dense and intricate vegetation representing ecological corridors necessary to the survival of many animal and plant species that survived the territory's intensive exploitation. Some of them are subjected to protection initiatives (e.g., Natura 2000 Network) [46].

In the Picone Torrent Basin, lithological terms of the Cretaceous carbonate succession of the Murgia emerge on which deposits of the Plio-Pleistocene coverage rest; there are also alluvial deposits from the Holocene age located on the bottom of the main erosive furrows.

Referring to the geological map of Italy in 1:100,000 scale [47] and to the geological map of the Murge and Salento in 1:250,000 scale [48], in the area of the Picone Torrent basin (Figure 2), the following lithostratigraphic units are recognized: Murge's limestones, Gravina's calcarenite, terraced marine deposits, and alluvial deposits.

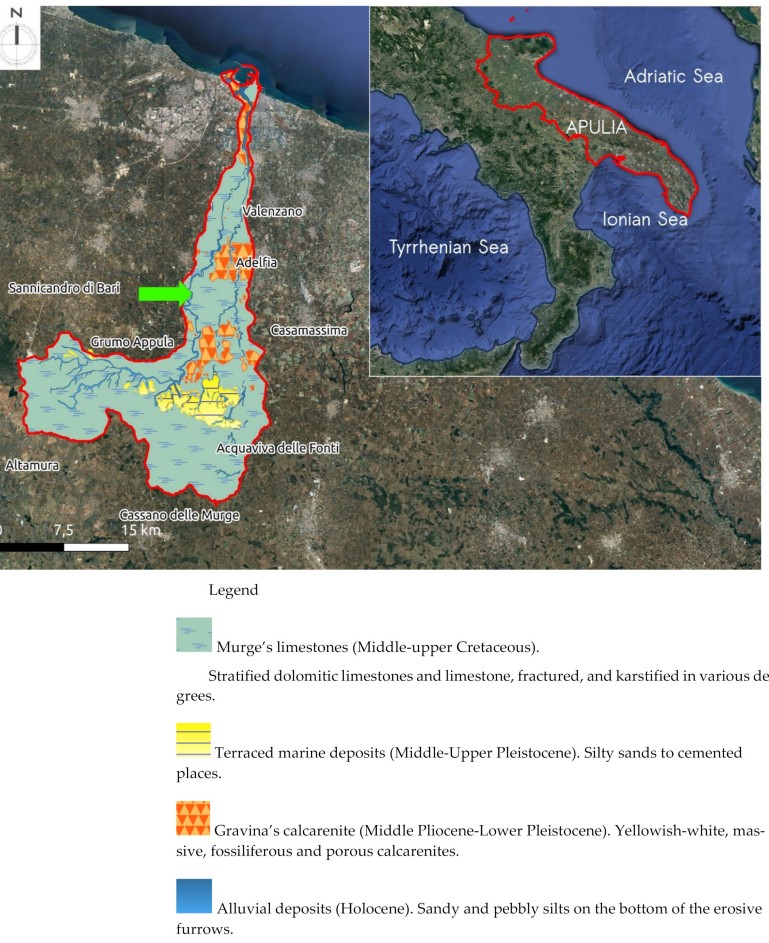

Legend

Murge's limestones (Middle-upper Cretaceous). Stratified dolomitic limestones and limestone, fractured, and karstified in various degrees.

Terraced marine deposits (Middle-Upper Pleistocene). Silty sands to cemented places.

Gravina's calcarenite (Middle Pliocene-Lower Pleistocene). Yellowish-white, massive, fossiliferous and porous calcarenites.

Alluvial deposits (Holocene). Sandy and pebbly silts on the bottom of the erosive furrows.

**Figure 2.** Location map of the study area (bordered by a red line) and geological map of the Picone Torrent Basin with lithostratigraphic units.

The Murge's limestones regroup the carbonate units as Bari' limestone and Altamura's limestone [49].

Bari's limestone constitutes the lower and middle part of the series and represents the oldest lithostratigraphy unit emerging in the area under study. It crops out in the central-northern part of the Picone stream basin and is formed by a succession of limestone, in layers or banks, micritic and granular, sometimes dolomitic, whitish, compact, and tenacious. Altamura limestone represents the upper part of the Cretaceous series of the

Murge Limestone Group. This unit is a succession of granular and micritic limestone, in layers or banks, characterized by abundant Rudiste remains (shells' fragments) with interspersed dolomites and grey dolomitic limestones [50,51].

Gravina's calcarenite crops start from the Picone stream basin's median part up to the closing section. The more significant extension outcrops are found between Adelfia and Sannicandro, showing a maximum thickness of 10 m, composed of limestones that are white-yellowish, coarse-grained, without stratification, and with numerous traces of bioturbations and fossil remains. On one hand, in some localities, macrofossils such as fragments or disarticulated valves of Pectinides and Echinides are found, and on the other hand, microfossils consisting of benthic foraminifera [48].

The terraced marine deposits emerge more extensively around 300 m of altitude, in correspondence to the towns of Cassano and Acquaviva. The terraced marine deposits rest on the limestone of the Murge and on the calcarenite of Gravina.

The stratigraphic succession is characterized by a karstified limestone-dolomitic substratum, on which sandy deposits rest with a level of thin greenish silt at the base. The sands, macrofossiliferous and fine-grained, show extremely variable densification and cementation; decimetric levels of clean, incoherent quartz sands are recognized alternating with highly cemented levels. Interposed between the Mesozoic substrate and the overlying Quaternary deposits, a discontinuous level of variable thickness (a few meters) of residual deposits (red earths) is found [52].

The alluvial deposits are Holocene deposits consisting of mainly calcareous pebbles and angular clasts deriving from the degradation and washout of the limestones by the meteoric waters that transport them and deposit them during alluvial events. The aquifer is found at gradually greater depths proceeding from the coastline toward the innermost areas and rests on the seawater of continental invasion; the karst aquifer circulates typically under pressure, below the altitude corresponding to the sea level, and is divided into several levels by sequences of more compact limestone strata [53].

### 2.1.1. Data Acquisition

In the present study, the GPR measurements were taken by a SIR 3000 Geophysical Survey System Instruments (GSSI) acquisition unit equipped with antennae with a central frequency of 400 and 200 MHz in the monostatic configuration. The scans were performed with the antenna assembled on a cart, equipped with a distance-measuring wheel encoder and transporting the instruments over parallel line scans. The trace-interval distance was about 4–5 m (the same as the olive planting pattern), with a time window respectively of 64 and 128 ns for the 400 MHz and the 200 MHz antennas. For both frequencies, 512 samples per scan were used. Dielectric was set to 15 (agricultural land, Table 1) and scan/unit (scans per unit of horizontal distance giving the resolution along survey line) to 18 because lower frequency antennas, like the 200 MHz, require coarser scan densities (having a smaller scan spacing produces higher resolution data, but larger file sizes). The number here is the number of scans that the system will collect per unit of distance).

Five parallel oriented SE–NW equally spaced reflection GPR profiles were surveyed. Three of the GPR profiles (nos. 3, 4, and 5, Figure 3) were placed within the suspected waste disposal area, whereas two of the profiles (nos. 1 and 2, Figure 3) were laid outside.

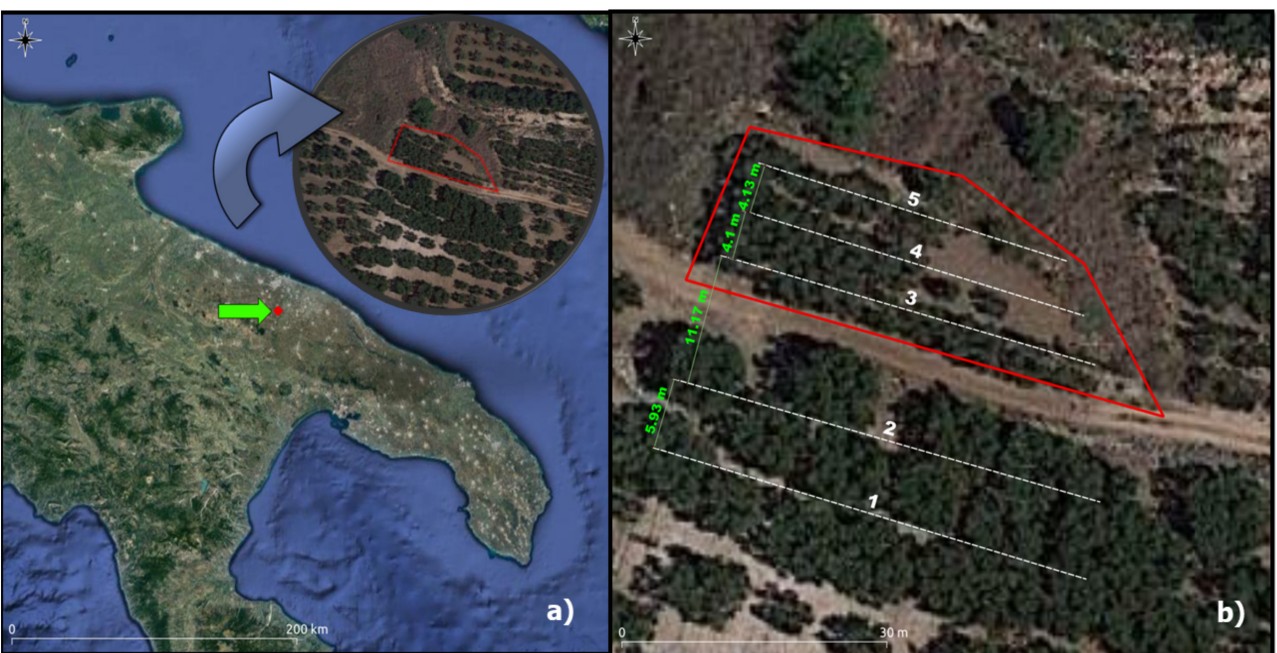

**Figure 3.** Study area (**a**) and scanning profiles (**b**) in the control (profiles nos. 1 and 2) and suspected area (profiles nos. 3, 4, and 5 in the red polygon).

The data were registered on a surface of about 1.500 m$^2$.

Trace scan length was from 26 to 30 m depending on the presence of Picone Torrent slopes. Interval among scans was approximatively 4 m due to the presence of olive tree rows (implanted after the artificial soil filling, probably to hide better the artificial buried materials), which hindered the ability to scan with the closest profiles (Figure 3).

Subsequently, calculating the ratio between the higher depth achievable and the better resolution reached by two different antennas, we present only the 400 MHz antenna results.

Profile no. 1 (Figures 3–5) represents the undisturbed soil below a centuries-old olive tree grove; this profile was acquired to emphasize and highlight the presence of natural rock bodies then disappeared in subsequent profiles. Profile no. 2 (Figures 3 and 6) was acquired on a path along with the shallow erosive furrows' side limit.

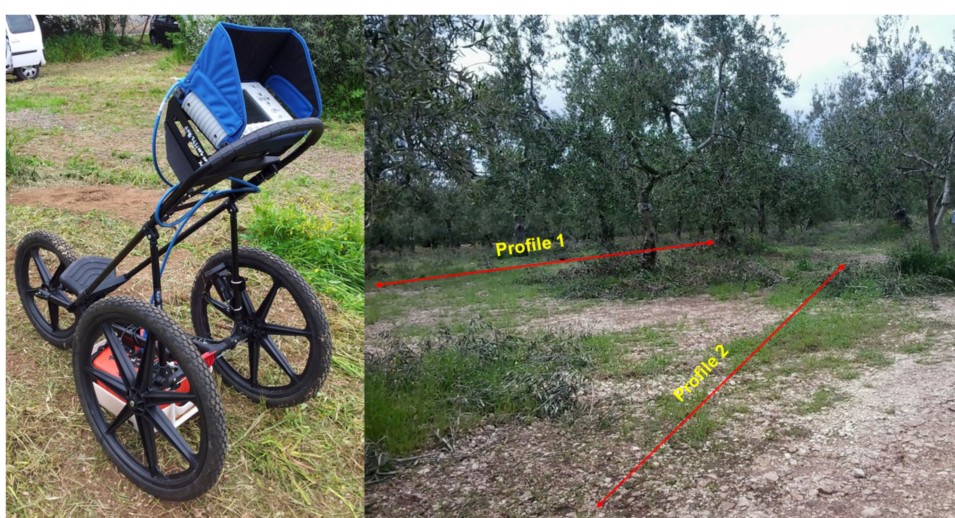

**Figure 4.** The image on the left represents the instrument configuration; on the right, profile nos. 1 and 2 acquired in the control area.

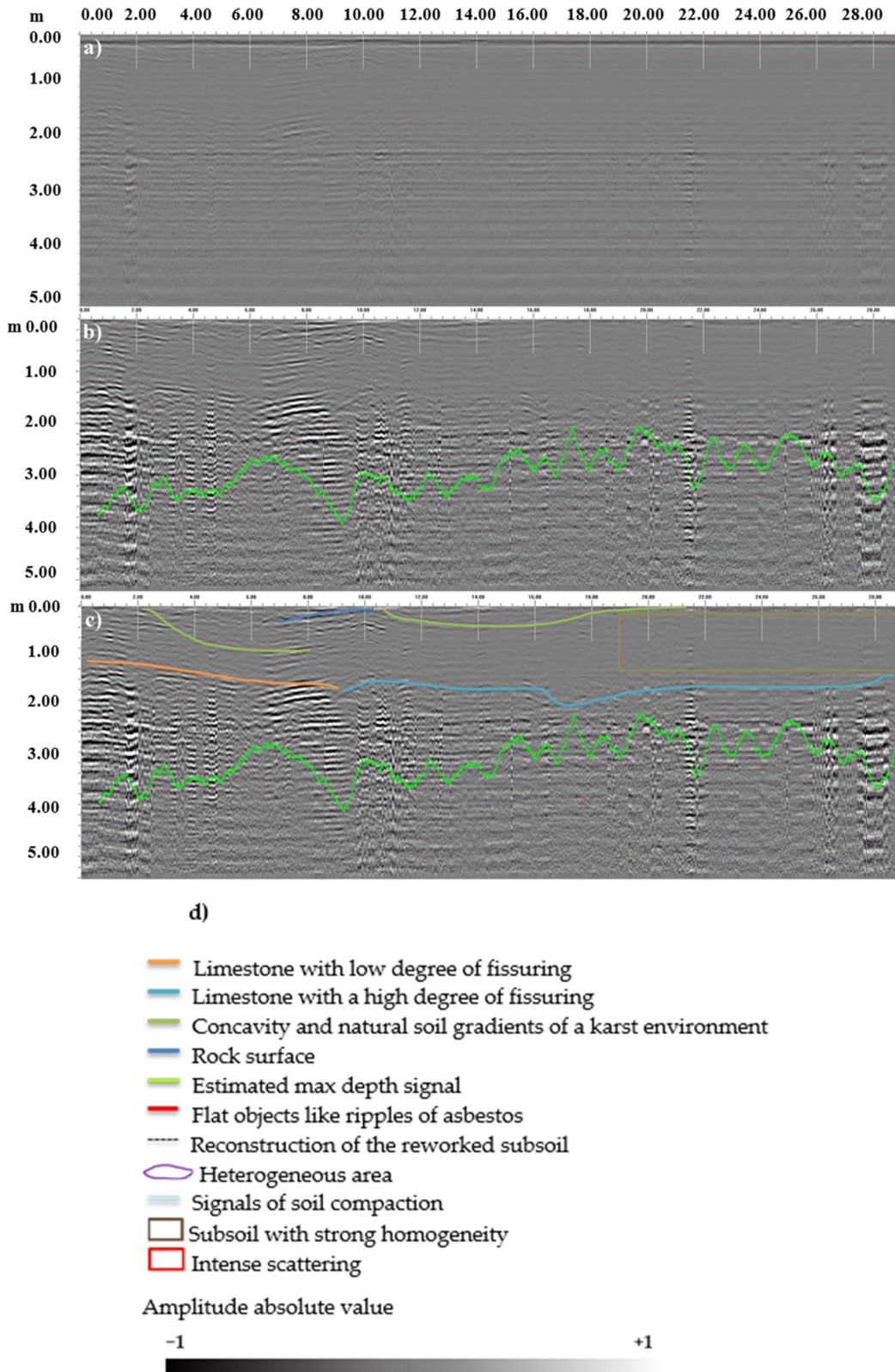

**Figure 5.** Line scan no. 1: (**a**) raw data; (**b**) processed data with estimated max depth signal; (**c**) interpreted data; (**d**) color legend.

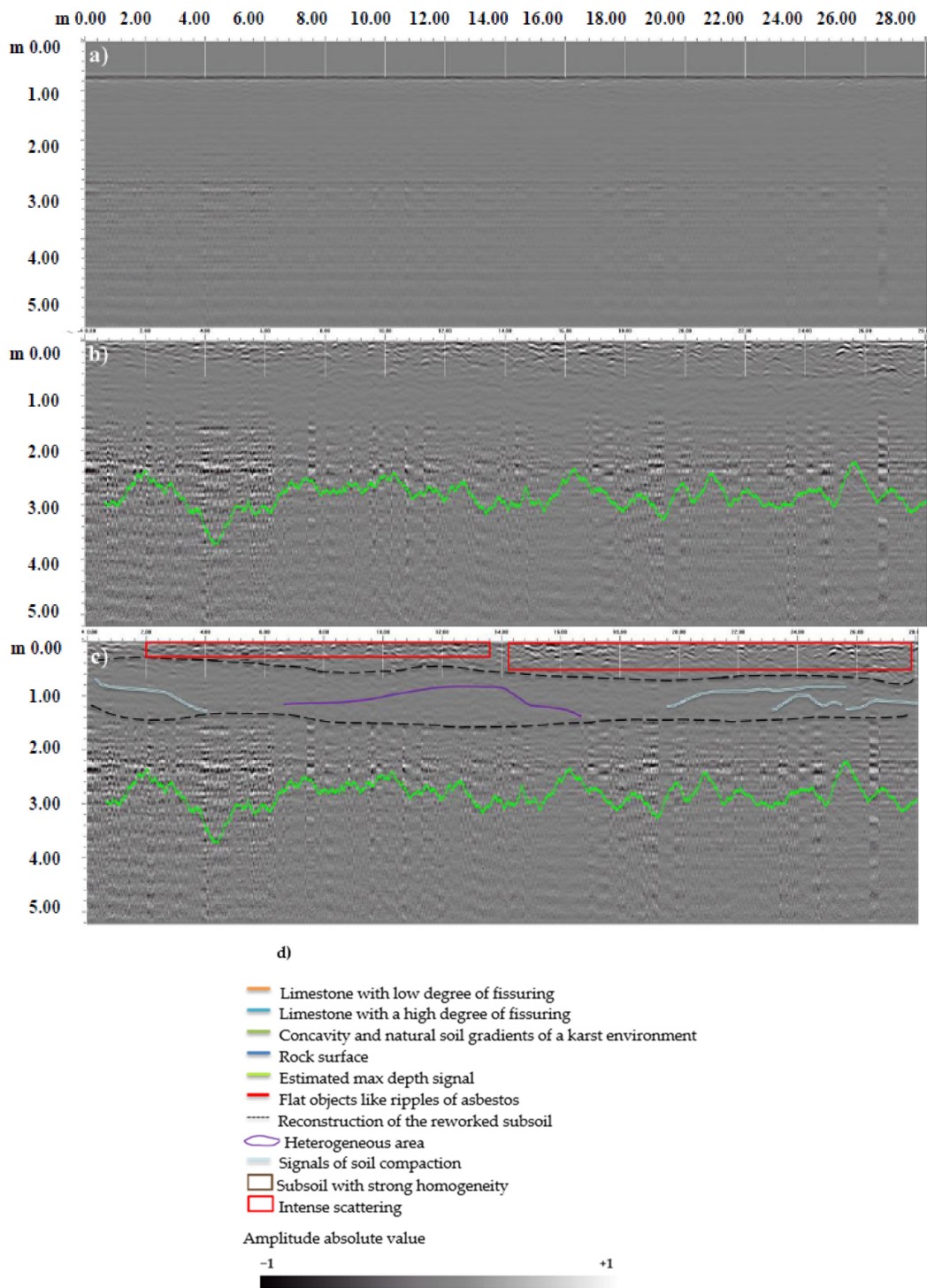

**Figure 6.** Line scan no. 2: (**a**) raw data; (**b**) processed data with estimated max depth signal; (**c**) interpreted data; (**d**) color legend.

Profiles no. 3 (Figures 3 and 7), 4 (Figures 3 and 8), and 5 (Figures 3 and 9) were acquired in the suspected area to reveal any anomalies.

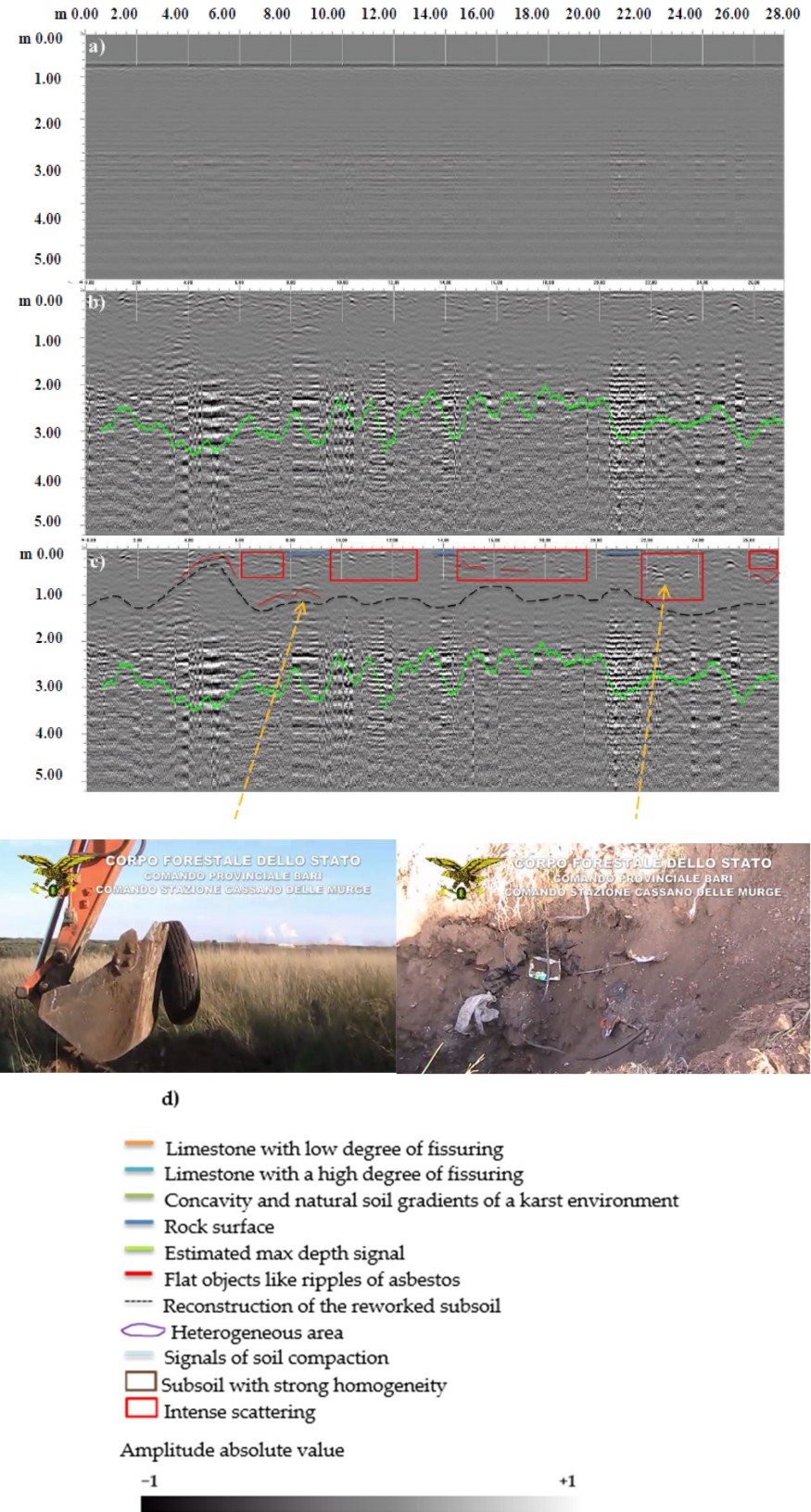

**Figure 7.** Line scan no. 3: (**a**) raw data; (**b**) processed data with estimated max depth signal; (**c**) interpreted data with pictures of some waste found during the excavation in the study site; (**d**) color legend.

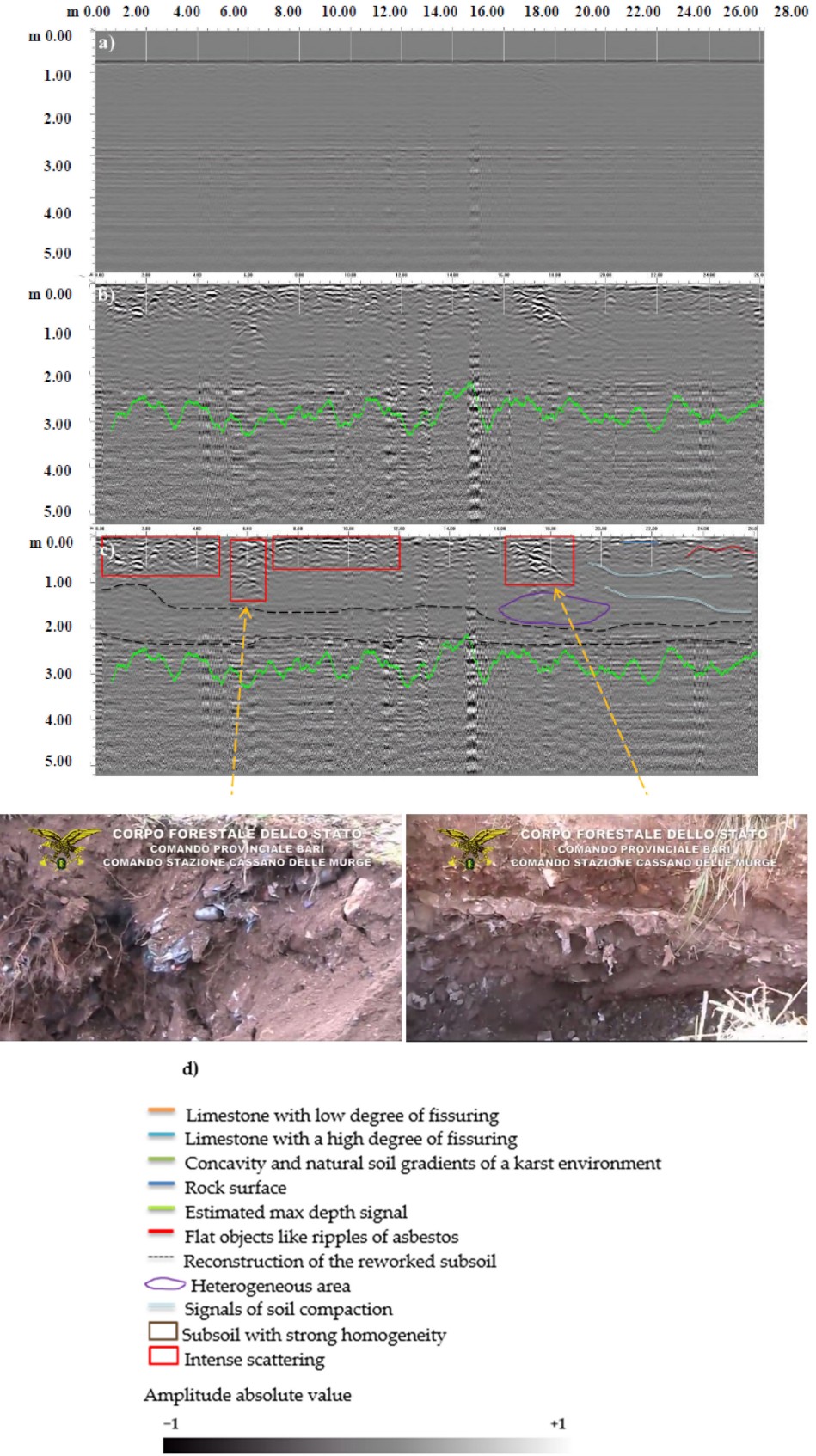

**Figure 8.** Line scan no. 4: (**a**) raw data; (**b**) processed data with estimated max depth signal; (**c**) interpreted data with pictures of some waste found during the excavation in the study site; (**d**) color legend

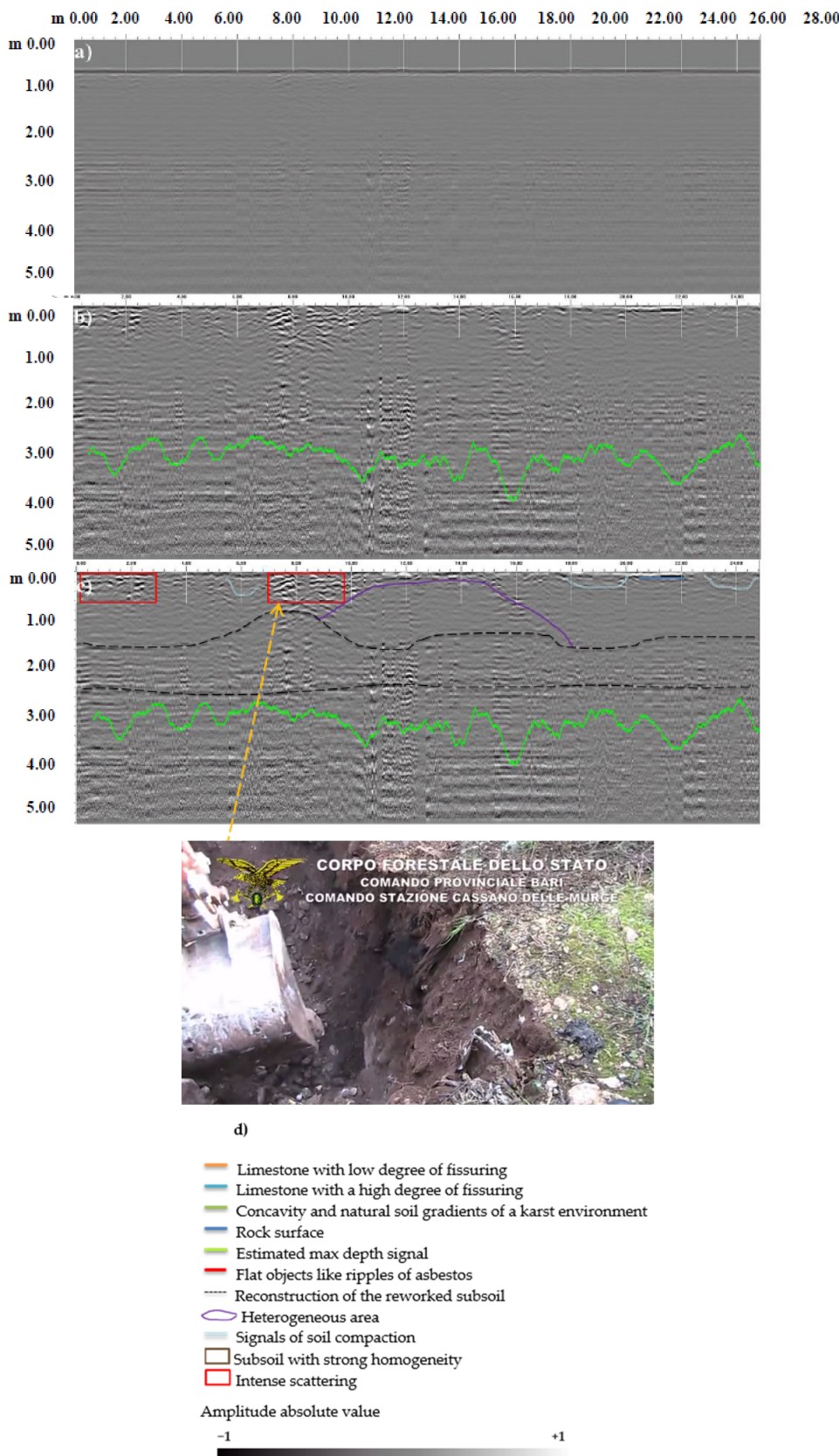

**Figure 9.** Line scan no. 5: (**a**) raw data; (**b**) processed data with estimated max depth signal; (**c**) interpreted data with pictures of some waste found during the excavation in the study site; (**d**) color legend.

Each profile is shown in Figures 5–8 three times as:

(a)　raw data;
(b)　processed data with identification through an analysis of the noise and signal loss maximum effective penetration;
(c)　identification and interpretation of the anomalies.

### 2.1.2. Data Processing

The acquired data were then processed using software RADAN 7 by adopting the following operative workflow:

1.　elevation correction is necessary to vertically adjust the whole profile's position in the data window (adjust time-zero). A corrected 0-position gave a better depth calculation as it sets the top of the scan to a more accurate approximation of the ground surface;
2.　background removal to suppress noise from the records is performed by calculating an average GPR trace using all the traces in the GPR section. Therefore, the average trace is subtracted to every single GPR trace, sample by sample [28,54];
3.　Infinite impulse response (IIR) application with an IIR Vertical High Pass Filter with a cut-off frequency corresponding to about 1⁄4 of the antenna's center frequency. Using a 400 MHz antenna, we set the high pass cut to 80 MHz as a starting point that gave us the best fit; and
4.　application of range gain option set to four points (the number of sections the data is divided equally vertically); gain corrections are applied to the entire dataset, where the best curve created will amplify the low amplitude signal and the high amplitude areas.

### 2.1.3. RDP (Relative Dielectric Permittivity) Estimation

As a useful tool in interpreting the signals acquired, the values of the antenna's footprint used at depths of 1 m, 2 m, 3 m, and 5 m with different RDP values (Table 1) referring to Equation (3) are shown in Table 2.

**Table 2.** Correlation between the central frequency of the antenna and the footprint at different depths.

| f | RDP | λ | A_1 | A_2 | A_3 | A_5 |
|---|---|---|---|---|---|---|
| MHz | - | m | m | m | m | m |
| | 1 | 0.75 | 0.89 | 1.60 | 2.31 | 3.72 |
| 400 | 5 | 0.34 | 0.49 | 0.90 | 1.31 | 2.13 |
| | 15 | 0.19 | 0.30 | 0.55 | 0.80 | 1.30 |
| | 25 | 0.15 | 0.23 | 0.43 | 0.63 | 1.02 |

f = central frequency, RDP = relative dielectric permittivity, λ = wavelength, A_1, A_2, A_3 and A_5= footprint at 1, 2, 3, 5 m depth.

## 3. Results and Discussion

The interpretation of the GPR profiles' results was generally supposed based on the local geology and the researchers' experience in field acquisitions.

Figures 5–9 show the results of the GPR profiles scanned.

In the radargrams related to profiles nos. 3, 4. and 5 (Figures 7–9), it is possible to differentiate areas of broad wave amplitudes (positive and negative) represented by alternating black and white bands and zones of tinier wave amplitudes (closer to zero), with a more constant shade of grey and a less distinguishable structure.

It was possible to interpret the signal qualitatively by a maximum depth of about 3 m (green lines in all Figure 5b,c, Figures 6, 7, 8 and 9b,c) using a function of the processing software that analyzes the noise and signal loss from scan to scan and provides an estimate of the effective depth penetration. The output of this function is shown as a green line on the screen.



The black dashed lines represent the limit identified by authors between anthropogenic material and natural soil. Here, the loss of the signal and the sharp decrease in wave amplitude is probably due to a compacted conductive layer located under the anthropogenic material that identifies a natural soil made of tiny particles that gave a wide water retention capacity and a larger electrical conductivity that heavily mitigates the propagation of radar waves.

The drought conditions registered before the acquisition day (24 April 2014), can further confirm the identified signal's maximum depth. In fact, the pluviometric data for the entire month of April indicate a number of monthly rainy days equal to one and a total of 4.2 mm of monthly rain [55].

Along profile no. 1, acquired outside the waste disposal site's limits, three different signal patterns recognized as superficial horizontal and subhorizontal reflectors were observed and underlined in Figure 5c by orange, light blue, and green lines. In the same profile, natural bodies like limestone with different degree of fissuring, concavities, and soils gradients, attributable to the signals observed, that disappeared in the other profiles were present. It is known that GPR is a helpful medium to reveal the presence of holes and underground caves [56–64]. Moreover, a relatively homogeneous layering of the subsurface reveals a peculiar horizontal stratification that shows noise and evident heterogeneity.

Similarly, in radargram no. 2 (Figure 6), a continuous scattering of the signal (red rectangles) along all scans in the first 20 cm evidenced a natural condition not encountered in the other profiles probably related to a non-stop layer of roots. Biomonitoring to evaluate tree root biomass and, consequently, forest productivity is a well-known GPR application [64]. Below these signals, high-signal attenuation is shown.

In the profile nos. 3, 4, and 5, where buried waste was suspected, the GPR response showed substantial differences not found in the control's profiles (nos. 1 and 2, Figure 4). In these profiles (nos. 3, 4, and 5), the data acquired identified anomalies related to the presence of discontinuous intense scattering regions (red rectangles) due to numerous anthropogenic bodies that caused diffraction hyperbolas.

In scan no. 4 (Figure 8), it is possible to observe a heterogeneous area (evidenced by an elliptical purple line) showing a flattening and compacting of layers of different composition, probably related to ancient contamination by waste now cemented with the inert ground, therefore, less electrically conductive, which did not mitigate the electromagnetic waves. Furthermore, alternations that began at the surface and extended over the whole depth (second red rectangle from the left) range corresponded to superficial litters and hide all targets existing below.

Similarly, in scan no. 5 (Figure 9), a dome structure was observed and its morphology combined with the disappearance of horizontal stratification suggests a structure not attributable to a natural cavity and that profound rearrangements occurred. In the same radargram, a high scattering region with low lateral continuity of reflections suggests that the waste was probably composed of building rubble.

As demonstrated in previous studies, electromagnetic wave scatters on crass debris such as construction waste will determine an important alteration in the reflected signal. Conversely, in the lack of such crass debris, the registered signal displays the same flat-lying pattern [65].

In these profiles, there were anomalies attributable to somewhat suspicious soil and subsoil movements and, in any case, not found in the profile comparison because they were really different if compared to those of an undisturbed and homogeneous soil (Figure 4) and thus probably due to illegal activities that caused rehashing and mixing the soil with waste.

The use of GPR allowed us to detect these anomalies in a very good way, as the scattering due to dome structures (purple lines) is related to the presence of cover layers of waste, which may be evident of the presence of clay nature materials (often used to retain odors) or to an excessive degree of soil compaction.

Such structures are well displayed in scan nos. 4 and 5 (Figures 8 and 9) as evidenced by heterogeneous areas and soil compaction signals.

These interpretations of radar profiles (nos. 3, 4, and 5) were confirmed with field excavations performed after completing the geophysical investigations to verify the nature of the anomalies revealed and clean up the site.

The excavation activity carried out on the following days found the hypotheses formulated following scans with GPR, bringing to light piles of materials containing asbestos, waste derived from construction and demolition activities, bituminous mixtures, and discarded tires as well as glass, plastic, and unsorted municipal waste.

Since the olive grove roots sank among the waste, investigations were then carried out to verify any critical repercussions on the study area's olive crops.

The landowners were reported for illegal landfill and violation of the legislation to protect the landscape.

## 4. Conclusions

The GPR method, used for our purpose, was demonstrated to be effective in detecting signal anomalies correlated to buried waste due to subsoil alterations caused by operations made to bury the waste, revealing promising implications in this field.

However, such inferences have to be interpreted carefully to exclude the anomalies that natural phenomena can explain. If, on one hand, it was possible to identify some of the signal anomalies attributable to the presence of buried waste; on the other hand, it was not possible to identify with certainty the nature of the buried material and to get a depth of investigation above that of 3 m. The interpretation of GPR data is one of the major challenges of this methodology and depends a lot on the method of acquiring and processing data and the field experience of the users.

Surely, to better comprehend GPR data, it is always necessary to compare the subsoil and the degree of homogeneity with a natural site where waste is not expected.

In conclusion, the GPR method stands out as being a tool in providing a continuous and powerful image of the internal subsurface structure, something even a high-density sampling scheme may still not be able to do.

**Author Contributions:** Conceptualization, C.C. and C.M.; methodology, C.M.; software, C.M.; validation, C.C. and V.F.U.; investigation, C.M. and V.F.U.; resources, V.F.U.; data curation, C.C.; writing—original draft preparation, C.C. and C.M.; writing—review and editing, C.C.; supervision, V.F.U. All authors have read and agreed to the published version of the manuscript.

**Funding:** This research received no external funding.

**Data Availability Statement:** Not applicable.

**Acknowledgments:** The authors wish to thank the State Forestry Corps for their willingness to publish the present field's survey results.

**Conflicts of Interest:** The authors declare no conflict of interest.

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
