# Peer review of "Ground Penetrating Radar as a Functional Tool to Outline the Presence of Buried Waste: A Case Study in South Italy"

_sustainability, doi:10.3390/su13073805_

Round 1

Reviewer 1 Report

This paper presents a somewhat basic case study. There is a lack of literature review and discussion, the methodological section should be improved  and the interpretation of the results is also quite basic and without a theoretical sound. Unfortunately, my recommendation is to reject and encourage authors to resubmit their work after some improvements. 

Attached you can find the PDF manuscript with my main criticisms and some recommendations. 

Reviewer 2 Report

see attached file

Reviewer 3 Report

In total a quite interesting paper presenting one of the common applications of GPR in the field of environmental geophysics. A very important fact for such studies that was correctly applied by the authors, is the comparison of profiles of undisturbed and disturbed soil to be able to conclude on buried waste. Furthermore, the validation of the geophysical results by excavations, like presented in the paper, is important.

Nevertheless some parts should be improved:

  1. You should mention more than 2 keywords.
  2. Your study is funded and well presented, but you are not the first reseachers doing work on this topic, so you should perhaps mention the results of other groups. E.g.:
    • Porsani et al. (2004): The use of GPR and VES in delineating a contamination plume in a landfill site: a case study in SE Brazil. - Journal of Applied Geophysics 55: pp. 199-209.
    • Jiang et al. (2012): Characteristics of Leakage Pollution of Longpan Road Gas Station and Its Enlightenment. – Journal of Environmental Protection 3: pp. 49-54.
  3. Please give your paper to a native English speaker to proof-read your language.
  4. Your Figure 1 is taken from another publication. Please cite it!
  5. Your section 2 is a long and comprehensive description of the local geology, but to further improve it, you should add a geological map to facilitate it for the reader to distinguish the different units.
  6. The upper 2 maps in Figure 2 have a low quality and the scale is too small to see the exact location. In the lower map, a background map/photo would be helpful to get an impression on the profile locations.

One principal remark: on p. 6 you mention that you used 18 scans/unit. This is quite coarse, as normally 50 scan/unit are used for a good resolution. Then you can even improve the signal/noise-ratio by a stacking of 3-4.

Reviewer 4 Report

Original Submission

Recommendation

Major revision

Comments to Author:

Title:  

Geophysical Surveys with the GPR Method to Detect Signal Anomalies Correlated to the Presence of Buried Waste. A Case Study.

Overview and general recommendation.

This paper is demonstrating a case study of GPR in Italy. This technique is very important and somehow accurate and has lots of important applications. First of all, as a person with more than 20 years familiarity with GPR, I like this paper very much; but as a scientist, I have to say the truth about the material and be honest. The title is somehow not satisfactory, I think you have to change it. The paper itself is a real chaos. As a reviewer I should reject it pronto, but I will do the review and we will see if it would be improved or not. English is a really disaster; you should show this to a native (that is a very very big must). Before uploading of the paper, you had to “accept all in the word processor”, so the output will seem much better. The abstract must be rewritten. It is more like an introduction. You have to give a bit of info and the unique results that you have gotten. Introduction is also a real mass: please make it neat sometimes you should combine the paragraphs. Refs are very bad, and must be given to anything that you say and you did not do. Section 1.1 is a disaster; please change it completely.

The formulas must be checked; I am sure lots of problems there. I am sorry to say that but you want to publish with MDPI and I am sure you know the gravity of the journal.

Sections 2.1.1 until 2.1.4 are copied from some articles (or geological maps)> you should not do that. All Figs must be managed better. Please look at the papers and see how they did that.

Conclusion also must be done again: it is very bad…

Lots of problems have been seen: please change the things that me and other reviewers said, after that I will check it again. I am sorry but we have to improve the text/material. This is a very good work, but, very badly written.

I have a question: how did you handle the “migration” on the data? I think the vertical resolution is very bad, and lots of filters must be done on the data (of course the data is from 2014 and it is impossible now. Am I right?).

Detailed comments:

line 21…. Above than (really?).

line 23. You have repeated GPR 2 times why? Please explain.

Line 26-28. I do not understand the meaning. Rewrite it again.

Line 29. Using word “several” 2 times? As I said, English must be checked.

Line 36…. Humidity content? It said water content!

Line 91-93. The most awful paragraph I have ever seen before. I am sorry but English must be checked as I said in the previous text.

Table.1 title: … velocity of what?

Line 122. R is reflection coefficient not amplitude. Please explain.

Line 153. Distance of what?

Fig.2: Make it neat. This figure in this shape cannot be accepted. Make a location map and show the study area on it. Legend is a disaster…

Line 181-184. You do not need to copy the legend of geology map and show us. You must be better than that.

Line 259: … 15 ms/m (give the dimension.

Fig.3. this Fig is also very bad: you could plot something better in GIS or other software.

Line 348: I think epsilon must be relative dielectric (am I right).

Fig.5. please explain how did you plot the colored lines and why? For instance, line green or others…

Round 2

Reviewer 1 Report

This reviewer acknowleges the effort made by authors. There are important gaps in the understanding of the GPR method. For example: 

Authors refer a trace-interval distance of 4 m. This is not correct and I think this is not the distance they have used. The distance between traces (A-Scans) is one of the most important parameters affecting image resolution. 

The scale of amplitud value should not be absolute (1/-1). It is important to know the magnitude of such intensity, and dielectric contrast, in orden to understand the probable targets.

The content of the manuscript should be better organized, mainly the methodological section. 
Introduction should provided more clear state of the art and to highlight The contribution and novelty of your work.

Author Response

Gentle reviewer thank you for comments and suggestions. 
We provided to amend and ameliorate the work considering all recommendations, and we tracked in blue all changes that we made into the text.

Authors refer a trace-interval distance of 4 m. This is not correct and I think this is not the distance they have used. The distance between traces (A-Scans) is one of the most important parameters affecting image resolution. 

the distance of the scans is shown through a GIS system.
We could not scan closer due to the presence of the trees.
However we did not do a 3D reconstruction of the subsoil,
therefore it was not necessary to acquire with greater
density.

The scale of amplitud value should not be absolute (1/-1). It is important to know the magnitude of such intensity, and dielectric contrast, in orden to understand the probable targets.

The absolute value is shown directly from the software
acquisition interface, so I couldn't give it a certain value.
The value of the dielectric constant, mentioned several times
in the text, is 15.

The content of the manuscript should be better organized, mainly the methodological section. 
Introduction should provided more clear state of the art and to highlight The contribution and novelty of your work.

Thanks for the suggestion, the mentioned sections have been
improved. The introduction was implemented by adding new
references, combining and moving the order of some paragraphs
and better explaining the purpose of the work and the
novelty.

Reviewer 2 Report

see attached file

Reviewer 4 Report

The authors have been responded almost all of the comments, and seems to me they are in good understating of the material; the paper is improved remarkably, and in this form, I am positive to accept that,

Good Luck!